# Bioactive Compounds and Their Chondroprotective Effects for Osteoarthritis Amelioration: A Focus on Nanotherapeutic Strategies, Epigenetic Modifications, and Gut Microbiota

**DOI:** 10.3390/nu16213587

**Published:** 2024-10-22

**Authors:** Kota Sri Naga Hridayanka, Asim K. Duttaroy, Sanjay Basak

**Affiliations:** 1Molecular Biology Division, National Institute of Nutrition, Indian Council of Medical Research, Hyderabad 500007, India; hridayanka@gmail.com; 2Department of Nutrition, Institute of Basic Medical Sciences, Faculty of Medicine, University of Oslo, 0317 Oslo, Norway; a.k.duttaroy@medisin.uio.no

**Keywords:** bioactive compound, polyphenol, nanocurcumin, epigenome, nanoformulation, non-coding RNA, gut microbiota, chondrocyte, osteoarthritis

## Abstract

In degenerative joint disease like osteoarthritis (OA), bioactive compounds like resveratrol, epigallocatechin gallate, curcumin, and other polyphenols often target various signalling pathways, including NFκB, TGFβ, and Wnt/β-catenin by executing epigenetic-modifying activities. Epigenetic modulation can target genes of disease pathophysiology via histone modification, promoter DNA methylation, and non-coding RNA expression, some of which are directly involved in OA but have been less explored. OA patients often seek options that can improve the quality of their life in addition to existing treatment with nonsteroidal anti-inflammatory drugs (NSAIDs). Although bioactive and natural compounds exhibit therapeutic potential against OA, several disadvantages loom, like insolubility and poor bioavailability. Nanoformulated bioactive compounds promise a better way to alleviate OA since they also control systemic events, including metabolic, immunological, and inflammatory responses, by modulating host gut microbiota that can regulate OA pathogenesis. Recent data suggest gut dysbiosis in OA. However, limited evidence is available on the role of bioactive compounds as epigenetic and gut modulators in ameliorating OA. Moreover, it is not known whether the effects of polyphenolic bioactive compounds on gut microbial response are mediated by epigenetic modulatory activities in OA. This narrative review highlights the nanotherapeutic strategies utilizing bioactive compounds, reporting their effects on chondrocyte growth, metabolism, and epigenetic modifications in osteoarthritis amelioration.

## 1. Introduction

Osteoarthritis (OA) is a progressive musculoskeletal disease resulting in pain, joint stiffness, and loss of flexibility caused by cartilage degradation and subchondral bone remodelling [1]. The prevalence of OA is rapidly increasing from 4.8% (1990) to 7.6% (2020) among the age-normalized global population [2]. In addition to affecting older people, OA also affects around 3.5% of working (30–60 years) adults, due to increasing rates of obesity and exposure to various lifestyle and environmental factors like diet, exercise, and others [2]. The OA pathogenesis involves narrowed joint space, bone marrow lesions, cartilage erosion, synovitis, and osteophyte formation [3]. Dysregulated intercellular cytokine-mediated communication resulting in inflammation is also involved in OA pathophysiology. Since inflammation and arthralgia are predominant symptoms of OA, non-steroidal anti-inflammatory drugs (NSAIDs) are often prescribed despite their side effects and toxicity. Chronic use of NSAIDs causes adverse drug reactions that can affect the cardiovascular, central nervous, renal, and gastrointestinal systems [4]. Other disease-modifying OA drugs include monoclonal antibodies, peptides, nucleic acids, enzyme inhibitors, cytokine inhibitors, and various bone-active growth factors. These have focused on inhibiting the progression of OA by preventing cartilage loss [5]. However, these drugs demonstrate a short half-life and cause adverse reactions, and thus are of limited therapeutic potential.

To date, OA is incurable, and joint replacement surgery is the last option to provide symptomatic relief. Despite surgery, joint function may not be restored entirely, and patients may experience flare-ups [5]. In addition, various post-surgery risks and complications and a high chance of revision surgery [6] have opened the scope for novel biotherapeutics. It has been estimated that roughly 47% of patients suffering from OA use alternate therapy, including lifestyle changes like exercise, diet, and nutraceuticals to alleviate pain [7]. NSAIDs commonly provide symptomatic relief by inhibiting cyclooxygenase and thereby preventing the biosynthesis of prostaglandins. However, inhibiting prostaglandin production might elevate the risk of renal, gastrointestinal, and cardiovascular complications [8,9]. Furthermore, reduced dosages of NSAIDs have been suggested, as they exert dose-dependent side effects. Several options, including nanoformulation and encapsulation, are emerging to enhance the therapeutic efficacy of the drugs at lower dosages [10]. These formulations have reduced particle size, improved bioavailability, and better absorption at the tissue level. Moreover, drug encapsulation limits systemic exposure, enhances drug solubility, and provides controlled release over time.

Several bioactive compounds, including polyphenols, have protective roles against inflammatory diseases, including OA [11]. Polyphenols play a role in modulating the immune system and are being studied from a therapeutic perspective [12]. Despite their potential, their therapeutic efficacies are limited by poor solubility and bioavailability, which can be improved by introducing encapsulation or using a nanocarrier delivery system [13]. Nanotherapeutic strategies comprise liposomes, micelles, polymers, and dendrimers, emerging as drug delivery systems carriers [14]. According to the European Commission, particle or particle aggregates in the size range of 1–100 nm are defined as nanomaterials. However, reference frameworks for their biosafety, quality, and efficacy are evolving to comply with the regulatory perspective of these formulations [15].

The use of nanotherapy in improving bone health, treating and preventing bone ailments, and maintaining a healthy bone status could have multiple benefits, but they need to be extensively reviewed. Despite emerging evidence of various polyphenolic bioactive compounds, including curcumin, as complementary therapy in addition to NSAIDs, the potential of nanotherapeutic strategies and their involvement in delaying the progression of OA is not known. Recently, epigenetic regulation of chondrocyte homeostasis during OA has drawn attention [16]. Understanding modalities in systemic responses involving gut microbiota and possible effects on epigenetic controls of OA pathogenesis could harness a viable therapeutic option in delaying and treating OA.

An extensive literature search was conducted using Scopus, PubMed, and Science Direct databases to find relevant studies using search terms like osteoarthritis, chondrocytes, inflammation, nanoformulation, curcumin, polyphenols, bioactive, epigenetics, gut microbiome, and similar keywords. Articles published in English over the past 15 years, comprising both clinical trials and pre-clinical studies, were included. This narrative review aims to summarize the current knowledge on potential mechanisms involved in the management of OA with the help of nanotherapeutic strategies.

## 2. Nanotherapeutics in Osteoarthritis

### 2.1. Nanoparticles as Carriers of Bioactive Compounds

Due to its avascular nature, cartilage tissue has a major limitation affecting its ability to heal or repair, which poses a challenge for drug delivery. Systemic delivery of drugs does not reduce disease symptoms, as the damaged tissue lacks vascularity. Hence, localized delivery of drugs via intra-articular administration could be a viable alternative [17]. The emergence of nanotherapeutics offers a potential benefit, due to various advantages in drug delivery for OA (Figure 1).

Compared with conventional therapy, nanotherapeutic strategies provide the advantages of improving drug delivery and efficacy, reducing cartilage damage, and promoting M2 macrophage polarization and extracellular matrix protein synthesis. Nanotherapeutics used in drug delivery can be classified based on the type of encapsulation carrier, such as polymer-based (polymer micelles, polymer nanoparticles, dendrimers, and polymer vesicles) or lipid-based (nanostructured lipid carriers, lipid nanocarriers, solid–lipid nanocarriers, liposomes, and nanoemulsions) [18]. Certain other inorganic nanoformulations have also been reported for use in OA. For example, a bifunctional controllable magnetothermal switch containing magnetic nanoparticles coupled with transient receptor potential vanilloid type 1 (TRPV1) antibodies was found to alleviate OA by preventing chondrocyte ferroptosis and macrophage inflammation via alternating magnetic field stimulation, attenuating synovitis and cartilage degeneration in mice [19]. Further, the limitations associated with the bioavailability of polyphenols as alternate therapy in OA have led to a shift in focus towards polyphenolic nanotherapeutics and drug delivery systems, which are widely gaining interest as methods for treating and slowing down disease progression [20].

Polymeric nanoformulations utilize natural or synthetic polymers that self-assemble to encapsulate drugs via emulsification, nanoprecipitation, and solvent evaporation techniques. These are emerging for their roles in drug delivery due to their inherent advantages like reduced toxicity, targeted site delivery, prolonged retention, controlled release, increased bioavailability, biodegradable properties, and increased drug solubility [21]. Polymers used in OA nanotherapy include chitosan, hyaluronic acid, polylactic acid (PLA), polylactic-co-glycolic acid (PLGA), polycaprolactone (PCL), and polyamidoamine (PAA). Chitosan is a cationic polymer that promotes transcellular and paracellular transport of drugs and exhibits pH-dependent drug release due to its ability to solubilize at acidic pH [22]. Hyaluronic acid is, in contrast, an anionic polymer present in the extracellular matrix of cartilage, with a specific binding affinity with the CD44 receptor [23]. PLGA is a synthetic biodegradable polymer that hydrolyses to form metabolite monomers including glycolic acid and lactic acid [24]. These polymers are finding applications both individually and synergistically for treating OA.

Lipid-based nanoformulation utilizes lipid biomolecules as a carrier and demonstrates superior targeted drug delivery in OA treatment via ligand–receptor interaction [25]. They comprise an aqueous region surrounded by a lipid bilayer where drugs are encapsulated and released gradually at the target site [18]. Liposomes are self-assembling spherical vesicles that elevate drug diffusion across the plasma membrane [26]. Celecoxib-loaded liposomes conjugated with hyaluronic acid hydrogels were found to be responsive to shear by restructuring, providing lubrication, increasing retention, and exhibiting targeted delivery of celecoxib (NSAID), thus attenuating OA [27]. Solid–lipid nanoparticles (SLNs) contain biocompatible lipids like biowaxes, fatty acids, or triglycerides that remain solid at room temperature. Solid–lipid nanoparticles conjugated with chondroitin sulfate and loaded with aceclofenac exhibited increased cellular uptake and extended drug release in an OA model [28]. Nanostructured lipid carriers (NLCs) are different from SLNs due to the use of liquid lipids, which confers their unique properties for drug delivery [29]. A topical gel containing NLCs loaded with ibuprofen enhanced drug permeability via the skin in mice, for treating joint inflammation associated with OA [30]. Nanoemulsions are liquid-in-liquid dispersions with increased stability, bioavailability, and cellular uptake [31]. Nanoemulsion gel of chondroitin sulfate and glucosamine alleviated knee OA by attenuating cartilage damage and pain [32].

### 2.2. Polyphenolic Nanotherapeutic and Osteoarthritis

The recent advancement of nanotechnology has significantly impacted disease treatment, particularly in the context of OA, by addressing the bioavailability challenges associated with bioactive polyphenols. Various nanotherapeutic strategies, including nanoemulsion, lipid nanoparticles, polymeric micelles, polymeric nanoparticles, organic-based nanoparticles, inorganic-based nanoparticles, carbon-based nanoparticles, β-lactoglobulin assembled nanoparticles, and metal oxide nanoparticles are currently under research [33]. When combined with bioactive compounds, these formulations enhance solubility, improve bioavailability, enable controlled release, and reduce the quantity and frequency of doses, thereby enhancing their therapeutic potential.

Plant polyphenols comprise multiple phenol units that occur naturally as secondary metabolites and have potent anti-inflammatory properties, as they act on pro-inflammatory mediators. In addition, polyphenols target oxidative stress by scavenging reactive-oxygen species (ROS) [34]. Recent data suggest that these polyphenolic compounds inhibit matrix degradation caused during OA pathogenesis by promoting the expression of extracellular matrix components such as aggrecan and type II collagen [35]. Some polyphenols also directly bind to matrix metalloproteinases (MMPs) and inhibit their activities [36]. Polyphenols also exhibit their chondroprotective role by activating the Nrf2/ARE pathway [37]. The Nrf2/ARE pathway has been identified as a key pathway in regulating cartilage degeneration via SOX-9 (SRY-box transcription factor 9) in an age-dependent manner [38]. This redox homeostasis signalling pathway attenuates chondrocyte apoptosis, oxidative stress, and extracellular matrix degradation [39]. Furthermore, polyphenol supplementation in knee OA patients improved physical function and alleviated pain and inflammation [40]. Various bioactive polyphenols such as methyl gallate, hydroxytyrosol, quercetin, and rosmarinic acid, which are possible therapeutics, showed chondroprotective effects to ameliorate OA [41,42]. Rosmarinic acid effectively suppressed MMP1, MMP3, and MMP13, subsequently upregulating type II collagen production [43,44].

The beneficial effects of bioactive compounds and their nanoformulation have only recently come to light. For instance, the poor bioavailability of hydroxytyrosol has led to the formulation of a hydrogel containing nanoparticles. These hydroxytyrosol–chitosan nanoparticles, in combination with hyaluronic acid and pluronic micelles, efficiently suppress pro-inflammatory effects and oxidative stress in chondrocytes [45]. Morin hydrate, a natural antioxidant and anti-inflammatory compound, has been used to formulate a copper–morin-based metal–organic framework (MOF) to mimic metalloenzymes [46]. These nanoenzymes suppress OA progression by repairing mitochondrial function and modifying the levels of pro-inflammatory markers. Nanofiber microspheres of tannic acid and strontium ions were found to reduce apoptosis, downregulate IL1β (interleukin 1β) and TNFα (tumor necrosis factor α) expression, and inhibit cartilage degradation upon intra-articular treatment [47]. Quercetin, curcumin, dimethyl curcumin, resveratrol, and oxymatrine are some bioactive compounds that have been encapsulated into liposomes and shown to modulate pro-inflammatory cytokines involved in OA [48]. Recent data show the beneficial effects of polyphenol nanoformulations in osteoarthritis (Table 1).

#### 2.2.1. Epigallocatechin Nanoformulations in Osteoarthritis

Epigallocatechin-3 gallate (EGCG) is a polyphenol, predominant in green tea and known for its antioxidant and anti-inflammatory effects [61]. In addition to its antioxidant properties, EGCG has been found to directly regulate signal transduction pathways, DNA methylation, and mitochondrial function by interacting with plasma proteins and phospholipids [62]. EGCG is a promising polyphenol for osteoarthritis due to its chondroprotective properties, and several preclinical and clinical studies have been reported [63]. Chondroprotective effects of EGCG have been observed in primary osteoarthritic chondrocytes. Several proteins such as IL6, MCP1, IL8, GM-CSF, GROα, GRO, MCP3, IP10, GCP2, NFκB, and NAP2 that were upregulated upon IL1β stimulation were found to be downregulated by EGCG [64]. EGCG has been reported to exhibit antiarthritic properties by epigenetic modulation of global miRNA expression, thus reducing inflammation in IL1β-induced primary chondrocytes. Furthermore, it was found that EGCG inhibited the expression of ADAMTS5 and COX2, which was upregulated upon IL1β stimulation via the modulation of hsa-miR-140-3p and hsa-miR-199a-3p [65,66]. ADAMTS5 upregulation leads to an early osteoarthritis event where aggrecan degradation occurs in the extracellular matrix [67]. Several nanotherapeutic strategies to improve the bioavailability of epigallocatechin-3 gallate have been researched. Copper EGCG nanosheets modulated ROS signaling and downregulated pro-inflammatory cytokines’ expression in primary chondrocytes by converting M1 macrophages to M2 macrophages [68]. An EGCG nanodrug in combination with selenomethionine significantly ameliorated OA by reducing oxidative stress and accumulation of Fe^2+^ and promoting activation of glutathione peroxidase 4 [69]. Nanoparticles of EGCG and glucosamine mixture displayed improved anti-inflammatory potential and higher antiarthritic activity compared with the native mix [70]. A novel formulation with a near-infrared nano-enzyme of EGCG with Au and Ag improved OA treatment by reducing cartilage damage [57]. Recent work on hydrogels of epigallocatechin-3 gallate seems to suggest a promising therapy for OA. EGCG in combination with hyaluronic acid showed chondroprotective effects by inhibiting IL1β, TNFα, and MMP13, scavenging ROS, and inducing M2 macrophage polarization in chondrocytes [71,72].

#### 2.2.2. Resveratrol Nanotherapeutic Strategies for Osteoarthritis

Resveratrol is a polyphenol known for its antioxidant and anti-inflammatory functions; it exerts chondroprotective effects by improving joint function and reducing cartilage degradation [73,74,75]. Limited clinical studies have reported resveratrol as a therapeutic in the treatment of osteoarthritis [76,77]. The poor solubility and low bioavailability of resveratrol have led to the exploration of its nanotherapeutic application. Resveratrol nano-encapsulated in a self-assembling lipid core led to reduced nitric oxide levels in primary human chondrocytes [78]. Resveratrol nanoemulsions were found to increase intracellular uptake compared with native resveratrol, minimizing toxicity and decreasing oxidative stress [79]. Furthermore, PLGA nanoparticles loaded with resveratrol were able to inhibit chondrocyte apoptosis, induce autophagy, and subsequently alleviate symptoms of OA [80]. Due to their sustained release, these nanoparticles showed improved bioavailability when used with quantifiable concentrations of resveratrol for 35 days. Differentially expressed genes such as CXCL1, IL6, NOX4, and MMP3 were upregulated in primary chondrocytes from OA patients and downregulated upon resveratrol treatment [81]. In addition, targeting the MALAT1/miR-9/NFκB1 and caspase-3/MMP13 axis with resveratrol revealed its therapeutic potential for managing OA [82]. Both these pathways regulate cellular homeostasis and inflammatory responses. The combined efficacy of resveratrol with meloxicam (NSAID) significantly alleviated symptoms of knee pain in OA and had superior efficacy compared with meloxicam alone [83,84]. The synergistic effect could have been due to the anti-inflammatory properties of resveratrol that aided in the drug’s action. A hydrogel of resveratrol combined with oxidised hyaluronic acid upregulated SOX9, aggrecan, and type II collagen while suppressing expression matrix metalloproteinases in LPS-induced inflammation in chondrocytes [85], indicating its further scope for eliminating the disadvantages of resveratrol’s bioaccessibility issue. It is also necessary to address the potential stability issues, toxicity, and poor loading capacity of resveratrol nanoformulations [86]. The clinical studies involving resveratrol in osteoarthritis are in their very early stages. More studies are required to establish optimal dosages. Although safety is not a concern since resveratrol is a plant polyphenol, dosage and efficacy need to be established.

#### 2.2.3. Nanoparticle-Based Therapies Involving Curcumin in Osteoarthritis

Curcumin, a dietary polyphenol and an active ingredient of *Curcuma longa*, has shown potent anti-inflammatory, anti-cancer, antioxidant, and anti-bacterial effects. Evidence suggests the anti-arthritic, immunomodulatory, and chondroprotective effects of curcumin [87]. Curcumin has emerged as a safer and more effective supplement for ameliorating OA [88]. Curcumin alone or co-supplemented with other polyphenols effectively reduced symptoms of OA, as evidenced by improved physical performance and reduced WOMAC (Western Ontario and McMaster Universities Osteoarthritis Index) and VAS (visual analog scale) scores [11]. Curcumin supplementation reduces the expression and secretion of various serum inflammatory markers such as IL4, IL6, TNFα, and CRP/hs-CRP (high-sensitivity C-reactive protein) [89,90,91,92]. Despite its potential, the poor solubility, limited distribution, restricted bioaccessibility, and inadequate bioavailability of curcumin restrict its therapeutic efficacy. Therefore, strategic nanotherapeutic approaches are extensively being adopted to improve the effectiveness of curcumin.

Curcumin nanoformulations have been designed using various methods to improve solubility, bioavailability, and targeted delivery. For example, self-assembling lipid-core curcumin nanocapsules showed improved solubility and were more effective in reducing nitric oxide-induced apoptosis in the inflamed chondrocytes [78]. Furthermore, to avoid any harmful effects of nanocarriers, a carrier-free assembly of curcumin and icariin nanoparticles by π–π stacking formulation exhibited improved cellular uptake and displayed prolonged drug release and synergistic anti-inflammatory effects, thus alleviating OA by protecting the cartilage [93]. Liposomal formulations of curcuminoids (curcumin and bisdemethoxycurcumin) promoted cellular uptake, improved the osteoprotegerin/receptor activator nuclear factor κB (OPG/RANKL) ratio and downregulated IL1β, demonstrating its role in attenuating OA progression and preventing osteoclastogenesis [94]. Curcumin’s antioxidant potential has been proven to increase upon its encapsulation [95]. To further enhance its antioxidant potential, a self-assembling ROS-responsive polymeric micelle encapsulated with curcumin displayed extended drug release and ROS scavenging when triggered with H_2_O_2_ in chondrocytes and in the OA rat model [96]. Polymeric micelles in a self-assembling acid-activatable curcumin polymer showed chondroprotective effects by downregulating IL1β and TNFα in the acidic microenvironment of monoiodoacetate (MIA)-induced OA [97,98]. Similarly, curcumin encapsulated in a pH-responsive cyclic brush zwitterionic polymer exhibited controlled drug release, upregulated type II collagen and aggrecan, suppressed pro-inflammatory markers such as MMP13 and IL1β, and improved lubrication in the synovial joints [99].

The efficacy of curcumin-loaded polylactic-co-glycolic acid nanoparticles (PLGA NPs) was tested in an MIA-induced OA model. They exhibited higher stability and bioavailability and greater chondroprotective effects compared with native molecules, by downregulating MMP1, MMP3, MMP13, IL1β, TNFα and improving locomotor function [100]. In another study, Cur-PLGA nanoparticles alleviated knee OA in rats by downregulating TNFα, IL6, IL1β, TGFβ, and NFκB [101]. An analysis of the synergistic effects of hyaluronic acid/chitosan nanoparticles and curcuminoids in the knee OA model revealed a decrease in chondrocyte apoptosis by upregulating the IκB and thereby inhibiting the NFκB pathway, downregulating MMP1 and MMP13 and promoting the expression of type II collagen [102]. A similar combination of hyaluronic acid and curcumin-loaded chitosan nanoparticles ameliorated OA, due to their prolonged retention in the synovial cavity, inhibiting chondrocyte apoptosis, downregulating pro-inflammatory cytokines, and upregulating RUNX2 and AP1 [103]. Curcumin microgel using poly (ethylene glycol) dimethacrylate efficiently attenuated IL1β-induced inflammatory response in chondrocytes and promoted cartilage repair in the OA model, thus displaying its pro-regenerative potential [104]. Curcumin was encapsulated into gelatin/silk fibroin microspheres to target localized delivery and studied for its effect on MIA-induced OA. Curcumin microspheres displayed extended anti-inflammatory effects, with significant histological improvements in rat bone tissue and reduced apoptosis and serum IL6 [105]. Another study involving silk fibroin nanoparticles (SFNs) demonstrated how curcumin’s efficacy and anti-inflammatory potential were enhanced when encapsulated into SFNs. Curcumin SFNs could provide an environment for controlled drug release and cytocompatibility, improve anti-inflammatory effects by regulating RANTES and IL6, and display antioxidant effects by ROS scavenging [106].

Several types of formulations are being developed to improve curcumin’s bioavailability further. For example, next-generation ultrasol curcumin (NGUC) efficiently alleviated OA pathophysiology by downregulating CRP, TNFα, IL6, IL1β, NFκB, COX2, and MMP3 and upregulating glutathione peroxidase (GPX), CAT, and SOD [107]. Another formulation, palmitoyl-glucosamine (PGA) co-micronized with curcumin, resulted in improved bioavailability and reduced pain, tissue damage, and paw edema in OA pathophysiology. The PGA-cur downregulated MIA-induced pro-inflammatory cytokines such as IL1β, TNFα, MMP1, MMP3, and MMP9 when integrated into the diet and was found to maintain meloxicam-based pain relief in dogs with OA pain [108,109]. A formulation of curcumin in water-dispersible form was found to facilitate MIA-induced OA by inhibiting chondrocyte apoptosis and cartilage damage, in addition to improving weight-bearing imbalance and downregulating caspase-3, nitrotyrosine, phospho-NFκB, and TNFα [110].

Several polyphenols including curcumin, resveratrol, and pine bark extract have been part of clinical interventions for knee osteoarthritis [92,111,112]. Despite this, clinical studies related to nanoparticle-based polyphenol in the context of osteoarthritis, which is the primary aim of this review, are limited. However, the few clinical studies that have been documented with curcumin are mentioned in Table 2.

Despite evidence of curcumin’s safety and efficacy, the mechanism of curcumin’s action in ameliorating OA is still unclear. Several mechanisms have been postulated, and curcumin might regulate more than one pathway simultaneously. Curcumin was found to modulate metabolism of key amino acids like threonine, serine, glycine, histidine, cysteine, methionine, glycerolipid, and inositol phosphate [120]. Curcumin also acts on the NFκB pathway, thereby inhibiting the expression of matrix metalloproteinases and reducing cartilage degradation [121]. Curcumin may act as a natural COX inhibitor and regulate inflammatory pathways [122]. Clinical trials with comparative supplementation of curcumin and an NSAID have evidenced similar efficacy [89,90,123]. Further, curcumin was shown to inhibit the production of pro-inflammatory mediators in an OA model in chondrocytes as well as in the articular cartilage of rats in vivo [124,125]. Despite the current knowledge gap regarding curcumin’s mode of action, nanoencapsulation and its use as a nutritional intervention have exhibited great potential.

## 3. Epigenetic Modulations in Osteoarthritis Pathogenesis

Gene expression modulates OA pathogenesis via epigenetic variables, including changes in chromatin structure, histone modifications, DNA methylation, transcriptional regulation, and the role of non-coding RNA. These mechanisms, in turn, modulate gene expression of transcription factors, specific extracellular matrix proteins, cytokines, and matrix-degrading proteinases [126]. Epigenetic regulation is also involved in dysregulated gene expression in the articular cartilage via the suppression of NFAT1 (nuclear factor of activated T cell 1) and SOX9 in an age-dependent manner [127].

### 3.1. Role of Histone Modifications in Osteoarthritis Pathology

Histone modifications and histone-modifying proteins play a significant role in cartilage development and, subsequently, OA disease pathology. Histone hyperacetylation has been linked to pathogenesis in rheumatoid arthritis patients [128]. p300/CBP, a histone acetyltransferase, is involved in the hyperacetylation of COL2A1 at H3K9 (lysine 9 on histone 3) and H4K8 (lysine 8 on histone 4) via SOX9 [129]. p300/CBP is associated with transcriptional factors, such as SOX5, SOX6, and SOX9, and upregulates cartilage oligomeric matrix protein [130]. Histone deacetylase (HDAC) enzymes have been indicated to play a role in regulating chondrocyte differentiation [131]. Zinc finger protein 521 (Zfp521) exerted chondroprotective functions by upregulating nuclear HDAC4 and promoting cartilage growth [132]. In osteoarthritic chondrocytes, upregulated HDAC1, HDAC2, and HDAC3 expression was observed [133,134]. Furthermore, HDAC7 overexpression upregulates MMP13 and impairs β-catenin and Wnt/β-catenin signalling in OA [135,136]. In this context, HDAC inhibitors (HDACis) that target Nrf2, MAPK, and NFκB can be potentially used as therapeutics for the management of OA [137]. EZH2 (enhancer of zeste homolog 2), a histone-lysine N-methyltransferase, upregulated SDC1 (syndecan-1) by inhibiting miR-138 via histone methylation in the promoter region and induced cartilage damage in IL1β-induced OA [138]. In osteoarthritic chondrocytes, EZH2 is significantly overexpressed, and upregulated Indian hedgehog (IHH), COL10A1, ADAMTS5, and MMP13 were downregulated upon EZH2 inhibition [139]. Further analysis revealed that EZH2 deficiency upregulated osteogenic gene expression in chondrocytes [140]. EZH2 inhibition could slow down OA progression by reducing cartilage degradation and downregulating the expression of genes associated with the inflammatory response [141]. In addition, NSD1, which is an H3K36 (36th lysine on histone 3) methyltransferase, was found to regulate cartilage homeostasis via upregulating OSR2 expression and H3K36 methylation [142]. Analysis of chromatin immunoprecipitation sequencing (ChIP-seq) data, RNA-seq data, and genome-wide association studies (GWAS) of OA using a 3D chromatin structure mapped onto a chondrocyte-specific network revealed possible risks in gene expression associated with OA [143]. GWAS data combined with Hi-C mapping led to the identification of genetic variants within chromatin loops, including enhancer–promoter loops, thus identifying putative effector genes such as PAPPA (pregnancy-associated plasma protein A) and SPRY4 (sprout RTK signalling antagonist 4) in OA [144].

### 3.2. Modulation of Promoter DNA Methylation in Osteoarthritis

Methylomes of healthy and OA cartilage show significant differences, with increased DNA methylation in the latter [145,146]. Increased promoter methylation has been negatively correlated with gene expression and vice versa [147]. Promoter-specific hypermethylation of COL9A1 and SOX9 has been evidenced in osteoarthritic chondrocytes [148,149]. In addition, promoter-specific hypomethylation of C-terminal binding protein (CtBP) was found to upregulate CtBP-dependent NLRP3 expression and downstream pathways and aggravate OA pathogenesis [150]. Studies have concluded that CpG sites on several genes such as MMP3, MMP9, MMP13, FGFR2, GDF5, SOST, RUNX2, IL1β, IL8, iNOS, COL11A2, COL9A3, and ADAMTS4, which play diverse roles in OA, undergo differential methylation [147]. Hypomethylation of leptin has been seen to upregulate leptin levels and MMP13, its downstream inhibitor contributing to OA progression [151]. DNA methylation studies to understand OA pathogenesis were initially limited to chondrocytes but are also being studied in synovial tissue and subchondral bone, where significant epigenetic changes have been noted [152,153]. Differentially methylated regions (DMRs) found in OA patients were characterized by increased inflammatory responses [154]. The majority of differentially methylated sites, around 50%, seen in the enhancer region could denote the critical role of enhancer methylation in OA [155]. DNA hypermethylation also suppresses miR146a and miR140-5p in osteoarthritic synoviocytes and chondrocytes, respectively [156].

The role of transcriptional regulation in the OA microenvironment and its involvement in disease pathogenesis has become increasingly apparent in recent years. Global gene expression analysis has revealed that certain transcription factors, like FOS, FOSL2, JUN, RELA, MYC, and EGR1 are suppressed in OA pathogenesis, with dysregulation of other genes involved in OA [157]. Furthermore, FoxO1 (forkhead box O family transcription factor) has been revealed to play a major role in OA pathophysiology by modulating autophagy and promoting proteoglycan expression. It may contribute to the upregulation of antioxidant genes [158,159,160]. As a therapeutic approach, a recent study found that histone deacetylase inhibitor panobinostat targets FoxO transcription factors, downregulates IL1β-induced pro-inflammatory cytokines, ameliorates OA by promoting autophagy, and reduces cartilage damage [161]. Connective tissue growth factor (CTGF) is also upregulated in cartilage surface damaged areas [162], which induces IL8 production [163]. Another transcription factor, SOX4, which is upregulated in OA, could be a potential therapeutic target in addition to its use as a diagnostic biomarker [164]. Growth differentiation factor 5 (GDF5) is vital in joint homeostasis and development. A functional polymorphism of GDF5 (rs143383) at 5′UTR has emerged as a significant risk factor for OA [165,166]. Overexpression of the transcription factor HOXA10 promoted GDF5 expression in a Gdf5-HiBiT knock-in mouse model [167]. These data suggest that modulation of promoter DNA methylation could play an intricate role in transcriptional regulation in OA pathogenesis.

### 3.3. Role of Non-Coding RNAs in Osteoarthritis Pathogenesis

Evidence elucidating the role of non-coding RNAs (ncRNAs) such as long non-coding RNA (lncRNA), microRNA (miRNA), circular RNA (circRNA), and small-interfering RNA (siRNA) in chondrogenesis, chondrocyte differentiation, and cartilage development in OA is expanding [168]. Further evidence suggests that crosstalk exists between the mRNA, lncRNA, circRNA, and miRNA that regulate OA progression [169]. Their specific roles in OA pathogenesis and associated therapeutic applications are currently being explored. Many studies have concluded that lncRNAs are involved in OA-related cartilage homeostasis [170]. LncRNAs such as MALAT1, HOTAIR, H19, XIST, and GAS5 have been extensively studied to determine their role in OA [171]. MALAT1 has been shown to promote OA-related matrix degradation and chondrocyte apoptosis by downregulating miR150-5p [172]. LncRNA-ATB downregulation in patients serves as a diagnostic marker for OA [173]. LncRNA SNHG1 overexpression in the OA model activates the PI3K/Akt pathway and reduces chondrocyte apoptosis [174]. LncRNA AC005165.1 mitigates inflammation and apoptosis in osteoarthritic chondrocytes by interacting with miR199a-3p and TXNIP [175]. Furthermore, L-glutamine was found to upregulate lncRNA NKILA, downregulate MMP13, NOS, COX2, NFκB, and TNFα, and subsequently reduce extracellular matrix degradation [176]. LncRNAs have also been linked to obesity, which is a significant risk factor for OA progression [177]. Some circRNAs such as circSOD2, circ_0044235, and cir_0037658 also regulate OA pathogenesis through modulating the growth-related signaling axis [178,179,180,181].

Aberrant expression of miRNAs has been associated with OA pathogenesis and progression. Increased expression of miRNA128a has been associated with cartilage degradation via repressing autophagy in OA chondrocytes [182]. Downregulation of miR214-3p expression activated NFκB signalling and aggravated OA [183]. Upregulation of miR103 expression in osteoarthritic cartilage also led to OA progression by inducing chondrocyte apoptosis [184]. Additionally, miR9-5p, miR146a-5p, miR138-5p, miR335-5p, and miR98 expression were also upregulated significantly in the osteoarthritic cartilage and serum compared with healthy controls and could serve as novel biomarkers [185,186]. Downregulation of miR222 expression led to the activation of HDAC4 and MMP13 and promoted cartilage degradation [187]. In addition to being involved in the progression of OA, miRNA expression is connected to several signaling pathways, including NFκB, IGF, TGFβ, and BMP. It regulates various proteins and cytokines involved in OA [188]. A recent meta-analysis of miRNA, SNP datasets, and differentially expressed mRNA in the osteoarthritic cartilage revealed novel OA-related genes such as LHDB, PPIB, ARCP4-TTLL3, TPI1, and ASS1, which could serve as therapeutic targets [189]. Integrated analysis of RNA-sequencing data obtained from healthy and OA subjects, microarray, and a gene expression dataset revealed that novel differentially expressed microRNAs (DEMs) such as miR4435-SOS1, miR4435-PIK3R3, miR584-5p-KRAS, and miR183-5p-NRAS and their target genes were involved in the primary OA regulatory network [190].

### 3.4. Polyphenols as a Potential Epigenetic Modulator

Polyphenols such as curcumin, resveratrol, EGCG, gallic acid, and genistein have been reported to modulate epigenetic changes in disease pathogenesis [191]. It has been found that autophagy could be dysregulated by altered epigenetic regulations whereby expression of several miRNAs is abnormally up- or downregulated in OA [192]. Autophagy, a mechanism to maintain cell homeostasis by intracellular degradation, is often dysregulated in OA. The ability of polyphenols to modulate epigenetic mechanisms can be exploited in this scenario to regulate autophagy and delay OA progression [193]. Among polyphenols, more data are emerging in relation to curcumin’s capacity to regulate inflammation and oxidative stress via epigenetic controlled transcriptional regulation and gene expression, as evidenced by pharmacokinetic and pharmacodynamic studies [194,195]. Curcumin inhibits p300 histone acetyl transferase (HAT) via proteosome-dependent degradation and displays anti-inflammatory effects by regulating TREM1, an amplifier of TLR-mediated responses [196,197]. Recent studies support curcumin-induced epigenetic changes by modulating miRNA expression, histone acetylation, deacetylation, and transcription factors [191,198].

RNA sequencing and differential gene expression analysis of *Curcuma longa* in primary chondrocytes have demonstrated significant modification of gene expression, leading to the downregulation of pro-inflammatory cytokines and upregulation of antioxidant and cytoprotective genes [199]. Several phytochemicals like berberine, resveratrol, curcumin, and EGCG have been shown to modulate the expression of lncRNA, and curcumin has been evidenced to modulate HOTAIR, PVT1, H19, LINC00623, and H2BFXP, which are also involved in OA [171,200]. Despite this, studies showing the direct effect of curcumin on OA by modulating long non-coding RNAs are limited. Curcumin has been studied for its role in modulating transcription factors—EGR1, NRF2, STATS, and PPARγ [201,202]. Curcumin was found to reverse diabetic nephropathy by activating FoxO3a transcription factor and NRF2 while inhibiting NFκB [203]. Curcumin was found to restore autophagy in foam cells by promoting the nuclear translocation of transcription factor EB (TFEB) and reducing inflammation [204]. Furthermore, tetrahydrocurcumin, a curcumin metabolite, could activate FoxO4, inhibit the PKB/Akt pathway, and attenuate oxidative stress [205] However, its potential mechanism in OA-related transcriptional regulation still needs to be established. Curcumin is also a modulator of DNA methyl transferases, but evidence concerning OA is lacking [206].

Concerning OA, curcumin demonstrated chondroprotective effects in high-fat diet-fed rats by reducing miR34a levels, which were associated with inhibition of apoptosis, upregulation of autophagy, upregulation of the E2F1/PITX1 pathway, and amelioration of OA-like lesions [207]. The administration of oral curcumin in addition to physical exercise, such as swimming, in the MIA-induced rat model led to decreased serum CRP and downregulated inflammatory cytokines. It restored HDAC3 and miR130a expression, easing joint stiffness and pain [208]. Curcumin primed onto extracellular vesicles (Cur-EVs) derived from bone marrow stem cells (BMSC) was tested on IL1β stimulated osteoarthritic chondrocytes. Cur-EVs could modulate inflammatory signaling by reducing apoptosis, PI3K/Akt phosphorylation, and alleviating IL1β-induced effects by upregulating miR126-3p in osteoarthritic chondrocytes, which was initially downregulated in the absence of Cur-EVs [209]. Further, curcumin-treated mesenchymal stem cells could modulate the upregulation of miR124 and miR143 by reducing their promoter-specific DNA methylation, thus attenuating OA progression [210]. Curcumin nanomicelles were found to suppress the expression of specific microRNAs such as miR16, miR138, and miR155 [117]. These miRNAs play crucial functions in immune responses during OA. Furthermore, this nanomicelle showed immunomodulatory effects by reducing the frequency of CD4^+^ T cells, CD8^+^ T cells, Th17 cells, and B cells while increasing the frequency of Treg cells [91]. RNA interference via siRNA-based therapy for OA is also steadily growing. For the delivery of curcumin and siRNA of hypoxia-inducible factor (HIF2α) into OA joints, a pH-responsive metal-organic framework was used for encapsulation. Synergistic efficacy of curcumin and HIF2α gene silencing led to reduced cartilage degradation and downregulated inflammatory response in IL1β-induced inflammatory chondrocytes [211]. Curcumin pre-treatment that targeted p300 HAT and STAT3 siRNA could synergistically effectively inhibit leptin-mediated IL8 in OA synovial fibroblasts [212].

## 4. Interplay of Gut Microbiome, Epigenetics, and Osteoarthritis: Mechanism of Bioactive Actions

### 4.1. Targeting Osteoarthritis by Epigenetic Modulation

Recent studies have elucidated the role of epigenetic regulation on gene expression involved in OA pathogenesis (Figure 2).

These studies not only aid in the understanding of OA pathogenesis but also serve as preceding evidence for future epigenetic-based therapeutic approaches [213,214]. Understanding the interactive perspective of the gut microbiome and epigenetics in OA pathophysiology provides precise direction targeting pathogenesis. While some miRNAs have been evidenced to be involved in disease progression of OA, as discussed in Section 3.3, others have exhibited their roles as therapeutic targets for OA. MicroRNA1 (miR1) was found to ameliorate cartilage damage by downregulating Indian hedgehog (IHH) signaling in a transgenic mouse model [215]. Overexpression of miR193b-5p downregulated HDAC7 and decreased MMP3 and MMP13 levels in an IL1β-induced OA model, demonstrating+ its protective role [216]. miR92a-3p and miR193b-3p expression increased promoter-specific histone acetylation on COL2A1, COMP, and AGGRECAN, and inhibited HDAC2 and HDAC3 expression, promoting chondrogenesis [134,217]. miR16-5p expression inhibited the MAPK pathway and ameliorated IL1β-induced OA in chondrocytes [218]. Overexpression of miR93 effectively suppressed TLR4 and NFκB signalling, thus downregulating the activation of various pro-inflammatory cytokines in chondrocytes [219]. MicroRNA31 overexpression led to upregulation of aggrecan and type I collagen and promoted chondrocyte proliferation [220]. Inhibiting miRNA126 led to decreased apoptosis and inflammation via upregulation of Bcl2 expression [221]. Again, inhibition of miR486-5p promoted aggrecan and type II collagen via the SMAD2 pathway in OA [222]. Recent studies have focused on nanoparticle-based delivery of miRNA therapeutics. For example, intra-articular delivery of miR141/200c antagomir in PEGylated polyamidoamine nanoparticles demonstrated increased retention and provided chondroprotection in transgenic mice [223]. Antagomir of miR365 encapsulated into nanotubes combined with yeast cell wall particles (YCWPs) quickly crossed the gastrointestinal tract after oral administration, showed no toxicity, inhibited target miRNA, downregulated several pro-inflammatory markers, decreased cartilage injury, and alleviated symptoms in an OA model [224].

Small interfering RNA (siRNA) utilizes the RNA interference (RNAi) mechanism to suppress target gene expression. siRNA-based therapeutics are being studied for their use in OA via targeting NFκB, mTORC1, TGFβ/SMAD, and Wnt/β-catenin pathways [225,226]. Epigenetic regulation of leptin via siRNA inhibited MMP13 expression in OA chondrocytes, thus exhibiting its therapeutic potential [151]. However, one major strength of siRNA therapy is its localized delivery at the target site. For this reason, siRNA-based nanotherapy is slowly gaining traction. siRNA nanotherapy can specifically target site-specific inflammatory genes, resulting in lesser pro-inflammatory effects and further improving tissue homeostasis and cartilage regeneration [227]. Nanocomplexes with suitable carriers are being developed to confer stability to siRNA and recognize avascular target cells in chondrocytes. For example, siRNA targeting MMP2 in complex with positively charged nanoparticles was demonstrated to counteract OA by preventing ECM degradation and cartilage damage [228]. siRNA nanocomplexes targeting MMP13 in the osteoarthritic model led to sustained release and reduced cartilage fibrillation and synovial inflammation [229,230]. Another combination of nitric oxide scavenger and siRNA targeting carbonic anhydrase 9 (CA-9)-NAHA-CaP/siCA9 nanoparticles alleviated OA progression by reducing pro-inflammatory effects and cartilage protection [231]. Furthermore, suppression of LPCAT3 by siRNA-lipid nanoparticles decreased OA-related cartilage erosion and pro-inflammatory cytokines via the MALAT1-LPCAT3 pathway [232].

### 4.2. Gut Microbiome-Mediated Epigenetic Modulation

The importance of diet and nutrition in health and disease and their possible modulation at the epigenetic level are increasingly being recognized in the field of nutriepigenetics. It is notable in this regard that the gut microbiome, which is influenced by diet, also modulates epigenetics right from early life [233]. The gut microbiota controls several systemic events, including metabolic, immunological, and inflammatory responses. The hypothesis of the gut–joint axis and its interaction with various factors like diet, physical exercise, age, and gender has been explored recently [234,235]. Recent studies have reported gut microbiome dysbiosis in OA [236,237]. Analysis of large-scale GWAS identified three major microbial taxa, *Methanobacteriaceae*, *Ruminiclostridium 5*, and *Desulfovibrionales*, that were associated with knee OA and showed protective effects associated with their prevalence [238]. Furthermore, an abundance of *Streptococcus* spp. resulted in an increased ratio of *Firmicutes*/*Bacteroides* and promoted local inflammation in OA [239]. Local inflammation due to this factor can lead to gut dysbiosis and reduce short chain fatty acid (SCFA) levels, thus posing a risk to the development of both obesity and OA. Obesity, metabolic syndrome, and their impact on the immune system are also major risk factors for OA progression [240]. In obesity, dietary factors and habits lead to aberrant epigenetic programming, resulting in hyperglycaemia and increased adiposity [241]. Obesity and gut dysbiosis cause chronic low-grade inflammation in chondrocytes, triggering inflammatory pathways that lead to cartilage degradation involving epigenetic alterations [242]. Prebiotics, probiotics, and synbiotics are beneficial in obesity-related OA [243]. These supplements might regulate gut metabolite synthesis and reduce OA disease progression. Gut microbiota have also been reported to influence the bioavailability and metabolism of OA drugs, as highlighted by the European Society for Clinical and Economic aspects of Osteoporosis, Osteoarthritis and Musculoskeletal diseases (ESCEO) [244].

The importance of bioactive compounds from the diet has also been elucidated, where gut metabolites derived from diet, mainly SCFAs, cause epigenetic changes [245]. Concerning obesity and related comorbidities, gut metabolites affect the host metabolism by epigenetic modulation of gluconeogenesis, lipogenesis, appetite, and inflammation, suggesting that diet can either positively or negatively affect the gut microbiota and associated metabolites [246]. Obesity-related gut dysbiosis also modulates inflammatory response via the epigenetic regulation of free fatty acid receptors (FFARs) [247]. In addition to SCFAs, other metabolites like vitamins, polyphenols, and polyamines cause epigenetic reprogramming [248,249]. Potential mechanisms for this epigenetic regulation include the action of gut metabolites on epigenetic enzyme regulators and substrates influencing enzyme activity of HDACs and DNMTs, involvement in DNA methylation, chromatin remodeling, histone modifications, and lncRNA and miRNA regulation [250,251]. Gut metabolites like butyrate, an SCFA that is commonly derived from *Roseburia* spp. or *Faecalibacterium prausnitzii*, have been shown to contribute to histone modifications [252]. Histone deacetylases modulated by butyrate and certain commensal bacteria are involved in histone crotonylation and maintain intestinal homeostasis [253,254]. The role of butyrate has recently been reported its association with disease-related gut dysbiosis. Butyric acid was shown to lower inflammation by regulating FOSL2 modifications [255]. Furthermore, a decrease in butyrate levels has been linked to overexpression of HDAC3, causing gut dysbiosis associated with diabetes [256]. A pilot study of whole-genome methylation analysis from blood and visceral adipose tissue revealed differential methylation levels in subjects with *Firmicute*-dominant bacteria, thus linking these with obesity and metabolic syndrome [257,258]. Increased abundance of *Fusobacterium* is potentially associated with DNA hypermethylation [259]. Commensal gut microbiota induces TET2/3-dependent localized DNA methylation during an inflammatory state and affects chromatin accessibility [260].

Inflammation is the major characteristic feature in OA pathogenesis. Gut microbiota as well as their derived metabolites have functions in immune modulation via epigenetic alterations. Lactic acid promoted acetylation of histone H3K27, inhibiting the pro-inflammatory function of macrophages and resulting in immunosuppression [261]. DNA methylation at CpG islands of Toll-like receptors such as TLR2 and TLR4 is linked to gut dysbiosis in terms of higher *Firmicutes*/*Bacteroidetes* ratio and abundance of lactic acid bacteria [262]. A randomized clinical trial reported that guided lifestyle changes and dietary interventions could decrease DNA methylation age (DNAmAge) by 3.23 years and potentially reverse epigenetic age in humans [263]. Indole-3-lactic acid derived from *Lactobacillus plantarum* has been reported to modulate chromatin accessibility in CD8^+^T cells [264]. Furthermore, β-glucan was found to inhibit the deposition of active histone marks upon LPS exposure [265]. Gut-derived inositol phosphate upregulates the activity of histone deacetylase-3 and promotes epithelial repair [266]. Capsiate, another gut metabolite, was found to inhibit ferroptosis involved in OA via the activation of SLC2A1 and HIF1α inhibition [267]. Thus, dietary modifications could reverse epigenetic changes, promote bone health, and slow disease progression by modulating gut microbiota.

## 5. Future Perspectives and Conclusions

Nutriepigenetics encodes the modulation of the epigenome by nutrients or bioactive compounds and represents an emerging area of research. Bioactive polyphenols can modify or reverse aberrant gene expression via epigenetics in OA pathologies [192]. Emerging studies that have examined the mechanisms of osteoarthritic and normal cartilage have reported that epigenetic deregulation is indeed involved in OA pathogenesis [268,269]. In synergy with pathogenesis, it is being revealed how bioactive compounds might act on the gut microbiome and alleviate OA via epigenetic modification. We recently gathered evidence that curcumin, one of the most studied bioactive compound, potentially modulates the gut microbiota and alleviates OA via the gut–bone axis [11]. Several studies have confirmed the increased efficacy of nanocurcumin over native, with an acceptable toxicity profile, biosafety, and chondroprotective effects [100,270,271]. Despite extensive research on this golden nutraceutical, the precise mechanisms involved in curcumin’s mode of action remain elusive. One reason may be its ability to produce pleiotropic effects by simultaneously modulating several signal transduction pathways and enzymatic activities, in addition to its newly established role in epigenetic regulation. Curcumin and its metabolites may positively modulate the gut microbiome and alleviate OA by altering intestinal barrier integrity [272], mucosal inflammation [272], gut microbial profile [273], and gut metabolites [274,275] and promoting chondroprotective and anti-inflammatory properties. Changes occurring at the gut–immune axis promote the ameliorating effects of curcumin against OA (Figure 3).

OA, while currently incurable, is witnessing a surge of potential treatments in the form of additional approaches including formulated dietary bioactive compounds. Several studies have reported that encapsulated nanoformulations exhibited superior effects in terms of bioavailability, therapeutic efficacy, polyphenol stability, targeted delivery, and sustained release [276,277]. These emerging approaches, facilitated by improved packaging and delivery to the target tissue, enhance the bioavailability of bioactive compounds or drugs. Nanoparticle-based therapies including bioactive compounds are showing promising benefits compared with native molecules, leading to higher therapeutic efficiency. These nanotherapeutic strategies, which include diverse types such as polymer-based micelles, dendrimers, vesicles, liposomes, and emulsions, demonstrate better solubility, stability, intracellular uptake, accessibility, bioavailability, and efficacy in the treatment of OA compared with native molecules.

Despite the several advantages of polyphenolic nanoformulations, certain limitations still exist that must be noted. The stability of the polyphenols is guaranteed in a nanoformulation as they are shielded; however, aggregation of these formulations could result in loss of functionality and might pose a threat in terms of toxicity. The right combination of bioactive compounds based on their compatibility in terms of various attributes, including solubility, efficacy, and toxicity, should be considered. Despite their potential in applications for drug delivery in OA, polymeric nanotherapeutics suffer from batch-to-batch variation, cytotoxicity, and issues of biosafety related to cationic polymers and phagocytosis by macrophages [21]. Nanoparticle-based approaches still present challenges, such as long-term bioavailability, safety concerns, and the need for large-scale clinical data relating to different ethnicities. Therefore, nanotherapeutic strategies should be optimized to obtain efficient nanocarriers for bioactive or drug delivery systems, ensuring stability, sustained release, prolonged bioavailability, increased cellular uptake, targeted delivery, and reduced adverse reactions when treating OA.

## Figures and Tables

**Figure 1 nutrients-16-03587-f001:**
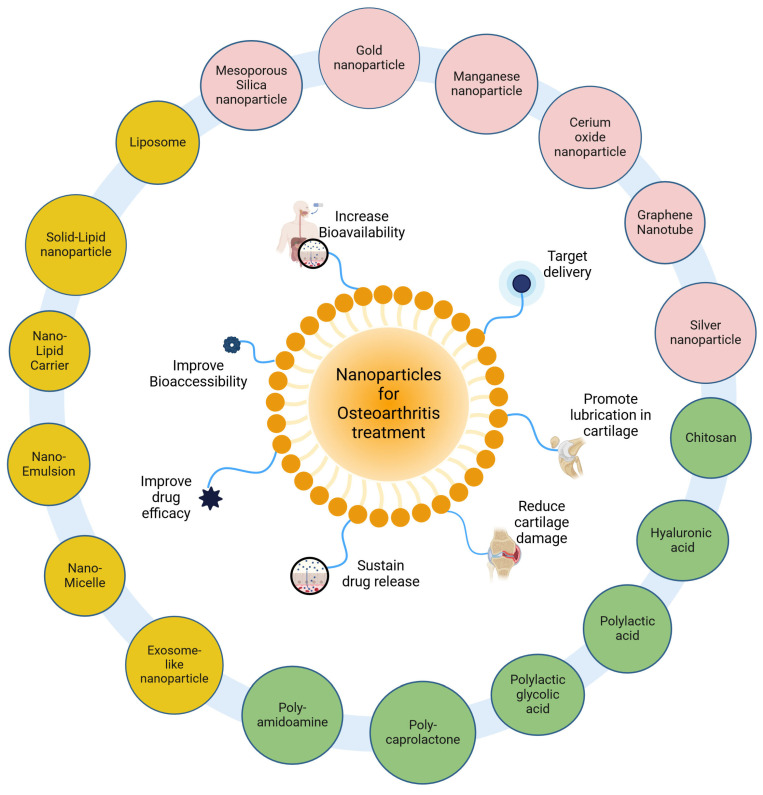
Nanotherapeutics in osteoarthritis. Nanoparticle-based therapy uses polymer, lipid, and inorganic materials, such as encapsulated carriers with drugs or bioactive compounds, to ameliorate osteoarthritis. A representative micelle structure (central) displays functional attributes of these nanoparticles. The yellow circle depicts a lipid-based nanocarrier, while the pink and green circles show inorganic and polymeric nanotherapeutic carriers.

**Figure 2 nutrients-16-03587-f002:**
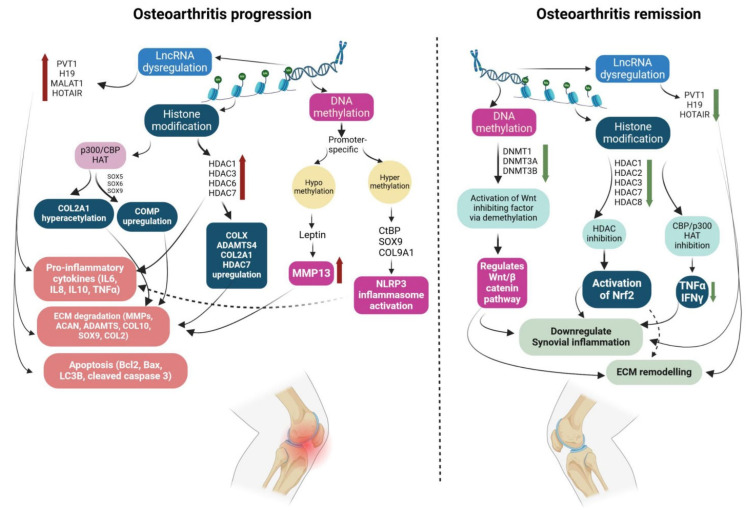
Epigenetic regulation in osteoarthritis. The role of epigenetic modulation in osteoarthritis is emerging and paving the way to new epigenetic targets for disease treatment. During osteoarthritis progression, modifications in histone acetyltransferase, histone deacetylase, DNA hypomethylation, DNA hypermethylation, and upregulated lncRNA collectively result in chondrocyte apoptosis, degradation of extracellular matrix, and activation of inflammasomes and inflammatory cytokines. Similarly, during disease remission, lncRNAs and DNA methyltransferases are downregulated, and certain histone deacetylases and histone acetyltransferases are inhibited, which subsequently downregulates cytokine activation and reduces cartilage damage by modulation of ECM proteins via several pathways. ECM—extracellular matrix; lncRNA—long non-coding RNA; PVT1—plasmacytoma variant translocation 1; MALAT1—metastasis-associated lung adenocarcinoma transcript 1; HOTAIR—HOX antisense intergenic RNA; HAT—histone acetyltransferase; HDAC—histone deacetylase; CBP—CREB binding protein; COL2A1—collagen type 2, alpha-1; COL9A1—collagen type 9, alpha-1; COL10/COLX—collagen 10; SOX9—SRY-box transcription factor 9; COMP—cartilage oligomeric matrix protein; CtBP—C-terminal binding protein; NLRP3—nucleotide-binding domain (NOD)-like receptor protein 3; IL6—interleukin 6; IL10—interleukin 10; TNFα—tumor necrosis factor alpha; IL8—interleukin 8; Bcl2—B-cell leukemia/lymphoma 2; Bax—Bcl-2 associated X protein; LC3B—light chain 3B; MMP—matrix metalloproteinases; ACAN—aggrecan; ADAMTS—A disintegrin and metalloproteinase with thrombospondin motif 5; DNMT—DNA methyltransferase; Nrf2—nuclear factor erythroid 2-related factor 2; IFN—interferon. Dotted arrow indicates possible pathway.

**Figure 3 nutrients-16-03587-f003:**
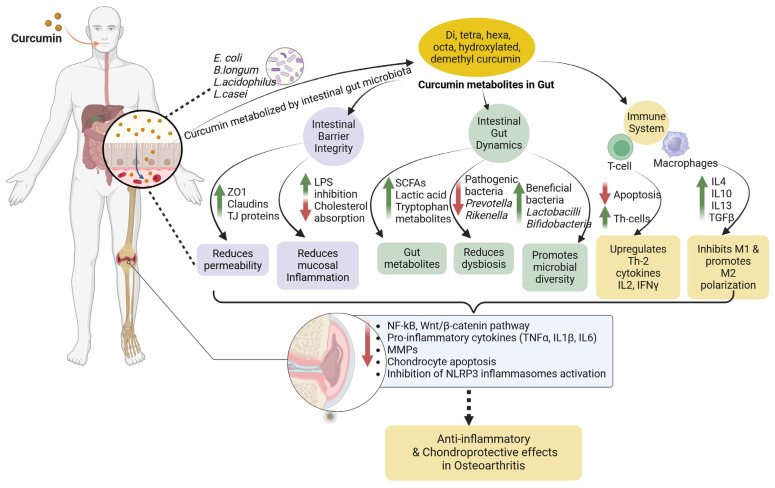
Potential therapeutic roles of curcumin in osteoarthritis via anti-inflammatory and chondroprotective effects. Curcumin, upon ingestion, undergoes metabolic reduction and is converted into various metabolites. Curcumin metabolites act on the gut–immune axis by promoting intestinal barrier integrity, immune function, and gut microbial diversity. It upregulates tight junction proteins and claudins and downregulates cholesterol absorption. Curcumin reduces gut dysbiosis and thus promotes the production of microbial metabolites like lactic acid and SCFA. Curcumin also promotes M2 macrophage polarization and regulates T-lymphocyte activity. These changes at the gut–immune axis promote the chondroprotective and anti-inflammatory effects of curcumin against osteoarthritis by inhibiting inflammasome activation, downregulating inflammatory cytokines, and reducing chondrocyte apoptosis. ZO1—zonula occludens-1; TJ—tight junction; LPS—lipopolysaccharide; SCFA—short chain fatty acid; IL2—interleukin 2; IFNγ—interferon gamma; IL4—interleukin-4; IL10—interleukin-10; IL13—interleukin-13; TGFβ—transforming growth factor beta; NF-kB—nuclear factor kappa B; TNFα—tumor necrosis factor alpha; IL1β—interleukin 1 beta; IL6—interleukin 6; MMP—matrix metalloproteinases; NLRP3—nucleotide-binding domain (NOD)-like receptor protein 3. Dotted arrow indicates possible pathway.

**Table 1 nutrients-16-03587-t001:** Evidence for the beneficial effects of polyphenol nanotherapeutic strategies in osteoarthritis.

Polyphenol Nanoformulation	Osteoarthritis Model	Effects In Vitro	Effects In Vivo	Refs.
Tannic acid/Sr^2+^-coated silk/graphene oxide-based meniscus scaffold	In vitro LPS-induced rabbit synovial MSC; In vivo papain-induced OA rat model.	Increased extracellular matrix secretion and promoted cell migration	Reduced cartilage degeneration and OA damage by downregulating MMP, IL6, IL8	[49]
pH responsive metal organic framework of hyaluronic acid loaded with protocatechuic acid	In vitro IL1β-induced rat primary chondrocytes; In vivo ALCT induced OA rat model.	Downregulated IL6, COX2, MMP1, MMP3, MMP13, ADAMTS5, and iNOS; Reduced synovial inflammation.	Promoted cartilage regeneration	[50]
Liposome-anchored teriparatide incorporated into gallic acid-grafted gelatin hydrogel	In vitro IL1β-induced mouse chondrocyte cell line, ATDC5; In vivo DMM-induced OA mouse model.	Activated expression of p-PI3K and p-AKT; Promoted anti-apoptotic effect by upregulating Bcl-2; Downregulated ADAMTS5.	- Upregulated expression of ACAN, SOX9; - Promoted glycosaminoglycan secretion and ROS scavenging	[51]
Gallic acid-encapsulated polymeric nanoliposome	In vitro H_2_O_2_-induced human chondrocyte cell line, C28/I2; In vivo MIA-induced OA rat model.	Promoted ROS scavenging; Lowered cartilage damage via upregulation of aggrecan and collagen II.	Mitigated joint wear by improving cartilage lubrication; Lowered cartilage erosion; Promoted ROS scavenging.	[52]
Gallic acid-loaded liposome with hyaluronan-grafted poly (2-acrylamide-2-methylpropanesulfonic acid sodium salt)	In vitro H_2_O_2_-induced human chondrocyte cell line, C28/I2. In vivo MIA-induced OA rat model.	Upregulated expression of Col II and ACAN; Reduced chondrocyte degeneration; Promoted antioxidant effect.	Lowered cartilage erosion and chondrocyte degeneration; Promoted glycosaminoglycan deposition.	[53]
Polydopamine-coated hesperetin-loaded Gd_2_(CO_3_)_3_ nanoparticles	In vitro IL1β-induced chondrocytes; In vivo ACLT-induced OA mouse model.	Downregulated TLR2, decreased inflammation, cellular apoptosis, and promoted chondrocyte maturation by inactivating NFκB/Akt pathway.	Displayed cartilage binding ability, mitigated cartilage degeneration.	[54]
pH-responsive polycaprolactone/polyethylene glycol naringenin nanofiber	In vitro IL1β-induced primary rat chondrocytes; In vivo ACLT-induced OA rat model.	Inhibited expression of IL6, IL1β, MMP3, and MMP13; Promoted COL2A1	Reduced cartilage damage; Increased proteoglycan retention and glycosaminoglycan content.	[55]
Scaffold of berberine–oleanolic acid complex grafted onto hyaluronic acid/silk fibroin composite	In vitro IL1β-induced OA model with primary rabbit articular chondrocytes	Upregulated COL1, COL2, and SOX9; Restored chondrocyte morphology	Promoted cartilage tissue regeneration in nude mice post-subcutaneous implantation.	[56]
NIR-responsive epigallocatechin gallate decorated Au-Ag nano-jars	In vitro H_2_O_2_-induced primary rat chondrocytes; In vivo ACLT-induced OA rat model.	Lowered chondrocyte apoptosis; Downregulated p-NFκB, iNOS, and COX2; Promoted cell migration	Reduced cartilage erosion; Improved cartilage thickness; Lowered chondrocyte apoptosis.	[57]
Hyaluronic acid-coated gelatin nanoparticles loaded with kaempferol	In vitro IL1β-induced primary rat chondrocytes; In vivo ACLT-induced OA rat model.	Downregulated expression of inflammatory cytokines COX2, MMP9, MMP13, TNFα, IL1β.	Attenuated inflammation and matrix degradation; Restored cartilage thickness.	[58]
Poly p-coumaric acid nanoparticles	In vivo temporomandibular joint OA rat model	--	Inhibited chondrocyte ferroptosis; Reduced ECM degradation; Exhibited long-term efficacy and alleviated cartilage repair.	[59]
Nano-naringenin	In vivo MIA-induced OA rat model	--	Upregulated GSH and TIMP3; Downregulated MDA, ADAMTS5 and MMP3.	[60]

Abbreviations: Sr^2+^—strontium ion; OA—osteoarthritis; MMP—matrix metalloproteinases; IL6—interleukin 6; IL8—interleukin 8; IL1β—interleukin 1 beta; ACLT—anterior cruciate ligament transection; COX2—cyclooxygenase 2; ADAMTS5—A disintegrin and metalloproteinase with thrombospondin motifs 5; iNOS—inducible nitric oxide synthase; ATDC5—mouse chondrogenic cell line; DMM—destabilization of medial meniscus; p-PI3K—phosphorylated phosphoinositide 3-kinase; p-AKT—phosphorylated protein kinase B; Bcl-2—B-cell lymphoma 2; ACAN—aggrecan; SOX9—SRY-box transcription factor 9; ROS—reactive oxygen species; H_2_O_2_—hydrogen peroxide; C28/I2—human chondrocyte cell line; MIA—monoiodoacetate; Col II—collagen type II; Gd_2_(CO_3_)_3_—gadolinium carbonate; TLR2—T-cell receptor 2; NFκB—nuclear factor kappa B; COL2A1—collagen type II alpha 1; COL—collagen; NIR—near-infrared; Au—gold; Ag—silver; TNFα—tumor necrosis factor alpha; ECM—extracellular matrix; GSH—glutathione; TIMP3—tissue inhibitor of metalloproteinase 3; MDA—malondialdehyde.

**Table 2 nutrients-16-03587-t002:** Effects of curcumin nanoformulations on osteoarthritis subjects: consolidated clinical trials.

Composition	Dosage and Delivery	Duration/Number of Subjects (*n*)	Key Outcomes	Refs.
Curcumin in water-dispersible form	180 mg/day, oral	8 weeks, *n* = 50	Reduced dependence on celecoxib vs. placebo group; lower VAS scores for knee pain.	[113]
Curcumin in water-dispersible form	180 mg/day, oral	6 months, *n* = 50	Improved JOA, VAS, and JKOM scores; no major side effects observed; 75.6% effective compared with placebo.	[114]
Curcumin in water-dispersible form	180 mg/day, oral	12 months, *n* = 50	Reduced cartilage stiffness and time-dependent decrease in scores of JOA, VAS, and JKOM.	[115]
Curcumin nanomicelle	80 mg/day, oral	6 weeks, *n* = 71	Significant decrease in WOMAC score.	[116]
Curcumin nanomicelle	80 mg/day, oral	3 months, *n* = 30	Decrease in VAS score; lower CRP levels; immunomodulatory effects on T cells and B cells.	[91]
Curcumin nanomicelle	80 mg/day, oral	3 months, *n* = 30	Suppressed expression of key miRNAs.	[117]
Solid lipid curcumin particles	160 mg/day, oral	3 months, *n* = 50	Improved WOMAC and VAS scores comparable to ibuprofen. No significant change was observed in inflammatory markers.	[118]
Curcumin self-nano-emulsifying-PEG organogel	1.5 g/twice per day, topical	8 weeks, *n* = 75	Significantly reduced WOMAC scores.	[119]

Abbreviations: VAS—visual analog scale; JOA—Japanese Orthopaedic Association; JKOM—Japanese Knee Osteoarthritis Measure; WOMAC—Western Ontario and McMaster Universities Osteoarthritis Index; CRP—C-reactive protein; PEG—polyethylene glycol.

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
