# Peer review of "Bioactive Compounds and Their Chondroprotective Effects for Osteoarthritis Amelioration: A Focus on Nanotherapeutic Strategies, Epigenetic Modifications, and Gut Microbiota"

_nutrients, 2024, doi:10.3390/nu16213587_

Round 1
Reviewer 1 Report
Comments and Suggestions for Authors
In the review entitled “Bioactive and its chondroprotective effects in ameliorating osteoarthritis: a focus on nanoparticle therapy” by Hridayanka and colleagues, the authors discuss how nanostructured formulations of bioactives show a greater potential in modulating metabolic, immunological and inflammatory responses, also involving the gut microbiota, which could play a role in the pathogenesis of OA.
The review is well written but some points could be revised.
Minor Point
Paragraph 2.2
Although paragraph 2.2 mentions the Nrf2/ARE pathway, the authors could further elaborate on the biological mechanism of how this pathway contributes to the protection of cartilage cells. A more detailed description would allow the reader to better understand the biological process. Additionally, it would be helpful to mention any limitations or challenges that still exist with the use of nanomedicine and polyphenols, such as safety concerns, long-term bioavailability, or lack of large-scale clinical data.
The authors could cite more recent studies regarding the efficacy of polyphenols in the clinical setting, providing updated data on the results obtained from more recent research, possibly with a separate section dedicated to ongoing or completed clinical studies. Finally, a final section summarizing the importance of nanomaterials and polyphenol combinations and future perspectives for research is missing. The authors could conclude with a clear and concise summary of the main advantages of these formulations and suggest areas of study that require further research.
Paragraph 2.3
In section 2.3, the authors could introduce epigallocatechin-3-gallate (EGCG) with a brief overview of its clinical and preclinical use, highlighting its relevance in the context of osteoarthritis (OA) and other pathologies. Furthermore, some of the described molecular mechanisms, such as the inhibition of ADAMTS5 and COX2 expression by modulation of specific miRNAs (hsa-miR-140-3p and hsa-miR-ù app199a-3p), could be better and more thoroughly explained. Finally, another insight into the role of these miRNAs in OA pathogenesis and how EGCG modulates these pathways would give the reader a more complete understanding of the cellular dynamics.
Paragraph 2.4
In section 2.4, some molecular mechanisms, such as the MALAT1/miR-9/NFκB1 axis and the caspase-3/MMP13 axis, are mentioned without in-depth explanations. The authors could expand on these points by describing how these mechanisms are involved in the pathogenesis of osteoarthritis and how resveratrol affects them. Additionally, while the text focuses on formulations that improve the bioavailability of resveratrol, there is no discussion of any limitations of these technologies. The authors could add information on any issues related to the production, stability, or side effects of resveratrol nanoformulations. Alongside this, the authors could include a short section on the most recent clinical studies involving resveratrol, describing which studies are ongoing or have been successfully completed. Including clinical data on efficacy, optimal dosages, and safety would improve the clinical relevance of the text. Finally, although the combination of resveratrol with meloxicam is mentioned, an explanation of why these combinations are promising is lacking. The authors could discuss in more detail the rationale behind the combination of resveratrol with nonsteroidal anti-inflammatory drugs (NSAIDs) such as meloxicam, clarifying how resveratrol can enhance the therapeutic effect and reduce the side effects of these drugs.
Author Response
Response to the Comments and Suggestions for Authors
Reviewer 1
Comment 1: In the review entitled “Bioactive and its chondroprotective effects in ameliorating osteoarthritis: a focus on nanoparticle therapy” by Hridayanka and colleagues, the authors discuss how nanostructured formulations of bioactives show a greater potential in modulating metabolic, immunological and inflammatory responses, also involving the gut microbiota, which could play a role in the pathogenesis of OA.
The review is well written but some points could be revised.
Response: Thank you for pointing out ways to improve the manuscript further. In the revised version, each comment is carefully reviewed and addressed as suggested.
Minor Point
Comment 2: Paragraph 2.2 : Although paragraph 2.2 mentions the Nrf2/ARE pathway, the authors could further elaborate on the biological mechanism of how this pathway contributes to the protection of cartilage cells. A more detailed description would allow the reader to better understand the biological process. Additionally, it would be helpful to mention any limitations or challenges that still exist with the use of nanomedicine and polyphenols, such as safety concerns, long-term bioavailability, or lack of large-scale clinical data.
Response: Thank you for your suggestion. In the revised version, we have elaborated sections with additional citations on the biological and regulatory mechanism of Nrf2/ARE pathway on cartilage remodelling. The following statements “Nrf2/ARE pathway has been identified as a key pathway in regulating cartilage degeneration via SOX-9 (SRY-box transcription factor 9) in an age-dependent manner [1]. This redox homeostasis signalling pathway attenuates chondrocyte apoptosis, oxidative stress, and extracellular matrix degradation [2]” has been added in LINE –174-176.
Limitations and challenges in regard to polyphenolic nanoformulations have been elaborated in the conclusion of the article (LINE – 621-625). The following statement “Nanoparticle-based approaches still present challenges, such as long-term bioavailability, safety concerns, and the need for large-scale clinical data with different ethnicities” is added as a limitation in the last paragraph of the conclusion.
Comment 3: The authors could cite more recent studies regarding the efficacy of polyphenols in the clinical setting, providing updated data on the results obtained from more recent research, possibly with a separate section dedicated to ongoing or completed clinical studies. Finally, a final section summarizing the importance of nanomaterials and polyphenol combinations and future perspectives for research is missing. The authors could conclude with a clear and concise summary of the main advantages of these formulations and suggest areas of study that require further research.
Response: Thank you for your suggestion. The present narrative review primarily emphasized nanoparticle-based polyphenolic studies in the context of Osteoarthritis. As such, several clinical studies have been carried out, and many are ongoing in the clinical setting involving polyphenols. In in our previous review, we have complied detailed clinical interventions with various bioactive supplements in knee osteoarthritis [3].
The potential efficacy of polyphenols in the clinical setting is mentioned in Line 306-309 as “Several polyphenols including curcumin, resveratrol and pine bark extract have been part of clinical interventions for knee osteoarthritis [4-6]. Despite this, in the context of osteoarthritis, clinical studies related to nanoparticle-based polyphenols are limited, which is the primary aim of this review. However, few clinical studies have been documented with curcumin, as mentioned in Table 2”.
A section summarizing the importance of nanomaterials and polyphenol combinations and future perspectives is added in Line 609-61 as “Several studies reported that encapsulated nanoformulations exhibited superior effects in terms of bioavailability, therapeutic efficacy, polyphenol stability, targeted delivery, and sustained release [7,8]”.
Comment 4: Paragraph 2.3 - In section 2.3, the authors could introduce epigallocatechin-3-gallate (EGCG) with a brief overview of its clinical and preclinical use, highlighting its relevance in the context of osteoarthritis (OA) and other pathologies. Furthermore, some of the described molecular mechanisms, such as the inhibition of ADAMTS5 and COX2 expression by modulation of specific miRNAs (hsa-miR-140-3p and hsa-miR-ù app199a-3p), could be better and more thoroughly explained. Finally, another insight into the role of these miRNAs in OA pathogenesis and how EGCG modulates these pathways would give the reader a more complete understanding of the cellular dynamics.
Response: Thank you for pointing this out. A statement is added to introduce epigallocatechin-3-gallate (EGCG) in Line 210-213 in section 2.2.1 as “In addition to their antioxidant properties, EGCG has been found to directly regulate signal transduction pathways, DNA methylation and mitochondrial function by interacting with plasma proteins and phospholipids [9]. EGCG is a promising polyphenol for osteoarthritis due to its chondroprotective properties, and several preclinical, and clinical studies have been reported [10]’’.
The molecular mechanisms involving ADAMTS5 are added in Line 219-220 as “ADAMTS5 upregulation leads to an early osteoarthritis event where aggrecan degradation occurs in the extracellular matrix [11]”.
Section 3.3 adds insight into the role of these miRNAs in OA pathogenesis, although limited information is available on how EGCG modulates miRNA in osteoarthritis.
Comment 5: Paragraph 2.4- In section 2.4, some molecular mechanisms, such as the MALAT1/miR-9/NFκB1 axis and the caspase-3/MMP13 axis, are mentioned without in-depth explanations. The authors could expand on these points by describing how these mechanisms are involved in the pathogenesis of osteoarthritis and how resveratrol affects them. Additionally, while the text focuses on formulations that improve the bioavailability of resveratrol, there is no discussion of any limitations of these technologies. The authors could add information on any issues related to the production, stability, or side effects of resveratrol nanoformulations. Alongside this, the authors could include a short section on the most recent clinical studies involving resveratrol, describing which studies are ongoing or have been successfully completed. Including clinical data on efficacy, optimal dosages, and safety would improve the clinical relevance of the text. Finally, although the combination of resveratrol with meloxicam is mentioned, an explanation of why these combinations are promising is lacking. The authors could discuss in more detail the rationale behind the combination of resveratrol with nonsteroidal anti-inflammatory drugs (NSAIDs) such as meloxicam, clarifying how resveratrol can enhance the therapeutic effect and reduce the side effects of these drugs.
Response: Thank you for your suggestion. We have revised the section with your valuable input and suggestions.
In the revised version, the following statements are added in this section as follows: LINE – 234- “Limited clinical studies have reported resveratrol as a therapeutic in the treatment of osteoarthritis [12,13]”
LINE – 243- “MALAT1/miR-9/NFκB1 axis and the caspase-3/MMP13 pathways regulate cellular homeostasis and inflammatory responses”.
LINE – 246 The synergistic effect could be due to the anti-inflammatory properties of resveratrol that aid in drug action.
LINE – 250- “It is also necessary to address and potential stability issues, toxicity and poor loading capacity of resveratrol nanoformulations [14]”.
Line 251 “The clinical studies involving resveratrol in osteoarthritis are in very early stages. More studies are required to establish optimal dosages. Although safety is not a concern since resveratrol is a plant polyphenol, dosage and efficacy need to be established”.
Reviewer 2 Report
Comments and Suggestions for Authors
This manuscript entitled “Bioactive and its chondroprotective effects in ameliorating osteoarthritis: a focus on nanoparticle therapy” included the research of English articles published in Scopus, PubMed, or Science Direct databases with terms of osteoarthritis, chondrocytes, inflammation, nanoformulation, curcumin, polyphenols, bioactive, epigenetics, gut microbiome, and/or similar keywords over the last 15 years comprising both clinical trials and pre-clinical studies and aimed to provide current knowledge of nanoformulated bioactives in controlling systemic events such as metabolic, immunological, and inflammatory responses and further modulating host gut microbiota to ameliorating osteoarthritis (OA). Herein, the interplays of bioactive compounds in gut microbiome and epigenetics were discussed intensively. This review is an emerging of nutriepigenetics that encodes the modulation of the epigenome by bioactive compounds or nutrients.
Overall, this review provides valuable information in the academic and therapeutic fields of OA disease. Importantly, it integrated current knowledges of bioactive, nanoparticle technology, epigenetics, and OA progression. In my opinion, it is suitable to be published in the “Nutrients”.
Author Response
Response: Thank you for accepting our work.
Reviewer 3 Report
Comments and Suggestions for Authors
The manuscript by Hridayanka, Duttaroy and Basak provides a summary of the recent scientific data on chondroprotective effects and OA ameliorating roles of bioactive compounds focusing on nanotechnological strategies for improved therapeutic index of bioactive compounds, epigenetic modifications and gut microbiota. The topic is interesting and complex. This review will be of benefit for the researchers working in the field. However, a thorough revision of the manuscript is necessary before publication due to the following points:
1. The topic of the manuscript includes more than three main directions – bioactive compounds for OA treatment, nanotechnological approaches for improvement of bioactive compounds effectiveness, epigenetic modifications and gut microbiota role in OA as well as their interplay with bioactive compounds with regard to OA pathology and disease amelioration. This multifactorial focus should be represented by the title of the manuscript. My suggestion for revision of the title is: “Bioactive compounds and their chondroprotective effects for osteoarthritis amelioration: a focus on nanotherapeutic strategies, epigenetic modifications and gut microbiota”.
2. The term “bioactive/bioactives” is more general. The article envisions mainly bioactive natural compounds, thus the authors should replace “bioactives” with bioactive compounds/bioactive natural compounds/bioactive substances (i.e. lines 2, 10, 18, 21, 22, 24, 27, 64, 78, 101, 177, 188, 517, 612, 655, 660, 662, 691, 693, 703, 708).
3. The authors should revise the last two sentences of the abstract.
a. L22-23: correction of English language is needed - … polyphenolic bioactive compounds modulate gut microbial response that result in …
b. L24-26: A more precise summary of the review focus should be presented (see point 1.).
4. Revision of the Introduction:
a. L37: replace “show” with involve.
b. L46: replace “these drugs” with they.
c. L49: replace “resort” with more appropriate word.
d. L69: Replace “Nanoformulation” with nanotherapeutic strategies. ( “Nanoformulation” is used in too many sentences of the text. It can be replaced with analogical terms in some parts.)
e. L76-79: the sentence should be revised in order to improve it and make it more clear to the reader.
f. L81: maybe the authors mean the instead of “their”?
g. L89: This narrative review aims to summarize current knowledge on potential mechanisms …
5. The authors should revise the title of subsection 2.1.: Nanoparticles and nanoemulsions as carriers of bioactive compounds.
6. L97: “boon” can be replaced with another word.
7. Figure 1 should be revised – the text in most of the outer circles is too small.
8. L112-115: The authors should explain more precisely the benefits of the mentioned modified magnetic nanoparticles. L131: delete “emerging”.
9. L138-140: Please, clarify the sentence.
10. The information on nanoemulsions in subsection 2.1 is scares. The authors should include more information if possible, if not – delete “nanoemulsions” in the title of the subsection.
11. L165: delete “molecules”. L176: include production – collagen type II production.
12. Subsections 2.3., 2.4. and 2.5. should be presented as subsubsections 2.2.1., 2.2.2., 2.2.3. and their titles should be revised, i.e. Epigallocatechin nanoformulations in OA, Resveratrol nanotherapeutic strategies for OA, Nanoparticle-based therapies involving curcumin in OA.
13. L236: PLGA nanoparticles loaded with resveratrol. L458: Include “s” – “Polyphenols”.
14. Some abbreviations are not defined first time they appear in the text (GPX, SDC1, etc.)
15. The authors should revise the title of subsection 4.1. – targeting OA by epigenetic modulation. L588-589: the definition for nutriepigenetics should be clarified.
16. L624-625: latin names should be italicized. L642: Toll-like receptors. L672: maybe the author mean gut-immune axis?
17. Why E. coli, B. longum, L.acidophillus and L. casei are shown as an example on Figure 3? It would be better to show main bacterial taxa involved in OA pathogenesis. Bigger font size is needed in the upper parts of the figure .
Comments on the Quality of English Language
Comments on the quality of English language are included in the previous comments.
Author Response
Reviewer 3
The manuscript by Hridayanka, Duttaroy and Basak provides a summary of the recent scientific data on chondroprotective effects and OA ameliorating roles of bioactive compounds focusing on nanotechnological strategies for improved therapeutic index of bioactive compounds, epigenetic modifications and gut microbiota. The topic is interesting and complex. This review will be of benefit for the researchers working in the field. However, a thorough revision of the manuscript is necessary before publication due to the following points:
- The topic of the manuscript includes more than three main directions – bioactive compounds for OA treatment, nanotechnological approaches for improvement of bioactive compounds effectiveness, epigenetic modifications and gut microbiota role in OA as well as their interplay with bioactive compounds with regard to OA pathology and disease amelioration. This multifactorial focus should be represented by the title of the manuscript. My suggestion for revision of the title is: “Bioactive compounds and their chondroprotective effects for osteoarthritis amelioration: a focus on nanotherapeutic strategies, epigenetic modifications and gut microbiota”.
Response: Thank you for your suggestion. We have revised the draft with your specific input and suggestions. The title has also been revised.
- The term “bioactive/bioactives” is more general. The article envisions mainly bioactive natural compounds, thus the authors should replace “bioactives” with bioactive compounds/bioactive natural compounds/bioactive substances (i.e. lines 2, 10, 18, 21, 22, 24, 27, 64, 78, 101, 177, 188, 517, 612, 655, 660, 662, 691, 693, 703, 708).
Response: We have replaced the term “bioactive” with “bioactive compounds” throughout the manuscript.
- The authors should revise the last two sentences of the abstract.
Response: We have revised the abstract.
- L22-23: correction of English language is needed - … polyphenolic bioactive compounds modulate gut microbial response that result in …
Response: We have revised the language.
- L24-26: A more precise summary of the review focus should be presented (see point 1.).
Response: We have revised the statement on precise summary of the review.
- Revision of the Introduction:
Response: The Introduction section is revised (Line 81-84)
- L37: replace “show” with involve.
Response: Replaced (line 41)
- L46: replace “these drugs” with they.
Response : Replaced (line 49)
- L49: replace “resort” with more appropriate word.
Response : Replaced with option (line 53)
- L69: Replace “Nanoformulation” with nanotherapeutic strategies. ( “Nanoformulation” is used in too many sentences of the text. It can be replaced with analogical terms in some parts.)
Response : The term “Nanoformulation” is replaced with either nanotherapeutic strategies or nanotherapeutic in several cases.
- L76-79: the sentence should be revised in order to improve it and make it more clear to the reader.
Response : The sentence is revised (line 81-83)
- L81: maybe the authors mean the instead of “their”?
Response : revised
- L89:This narrative review aims to summarize current knowledge on potential mechanisms …
Response : The sentence is revised (line 96-98)
- The authors should revise the title of subsection 2.1.: Nanoparticles and nanoemulsions as carriers of bioactive compounds.
Response : The title of subsection 2.1 is revised
- L97: “boon” can be replaced with another word.
Response : replaced with potential benefit (line 108)
- Figure 1 should be revised – the text in most of the outer circles is too small.
Response : The text in most of the outer circles is now made bigger as far as possible considering the limitation of the outer frame size.
- L112-115: The authors should explain more precisely the benefits of the mentioned modified magnetic nanoparticles. L131: delete “emerging”.
Response : The section is revised (line 126-129)
- L138-140: Please, clarify the sentence.
Response : The section is clarified (line 153-154)
- The information on nanoemulsions in subsection 2.1 is scares. The authors should include more information if possible, if not – delete “nanoemulsions” in the title of the subsection.
Response : “nanoemulsions” is deleted
- L165: delete “molecules”. L176: include production – collagen type II production.
Response : Revised
- Subsections 2.3., 2.4. and 2.5. should be presented as subsubsections 2.2.1., 2.2.2., 2.2.3. and their titles should be revised, i.e. Epigallocatechin nanoformulations in OA, Resveratrol nanotherapeutic strategies for OA, Nanoparticle-based therapies involving curcumin in OA.
Response : The section is revised as per the suggestions
- L236: PLGA nanoparticles loaded with resveratrol. L458: Include “s” – “Polyphenols”.
Response : Corrected
- Some abbreviations are not defined first time they appear in the text (GPX, SDC1, etc.)
Response : These abbreviations are elaborated.
- The authors should revise the title of subsection 4.1. – targeting OA by epigenetic modulation. L588-589: the definition for nutriepigenetics should be clarified.
Response : The section is revised as per the suggestions
- L624-625: latin names should be italicized. L642: Toll-like receptors. L672: maybe the author mean gut-immune axis?
Response : Yes, it is axis
- Why E. coli, B. longum, L.acidophillus and L. casei are shown as an example on Figure 3? It would be better to show main bacterial taxa involved in OA pathogenesis. Bigger font size is needed in the upper parts of the figure.
Response: The bacterial strains have been mentioned as they have been specifically identified to metabolize curcumin [15].
References
- Kubo, Y.; Beckmann, R.; Fragoulis, A.; Conrads, C.; Pavanram, P.; Nebelung, S.; Wolf, M.; Wruck, C.J.; Jahr, H.; Pufe, T. Nrf2/ARE Signaling Directly Regulates SOX9 to Potentially Alter Age-Dependent Cartilage Degeneration. Antioxidants (Basel, Switzerland) 2022, 11, doi:10.3390/antiox11020263.
- Khan, N.M.; Ahmad, I.; Haqqi, T.M. Nrf2/ARE pathway attenuates oxidative and apoptotic response in human osteoarthritis chondrocytes by activating ERK1/2/ELK1-P70S6K-P90RSK signaling axis. Free radical biology & medicine 2018, 116, 159-171, doi:10.1016/j.freeradbiomed.2018.01.013.
- Basak, S.; Hridayanka, K.S.N.; Duttaroy, A.K. Bioactives and their roles in bone metabolism of osteoarthritis: evidence and mechanisms on gut-bone axis. Front Immunol 2024, 14, 1323233, doi:10.3389/fimmu.2023.1323233.
- Mülek, M.; Seefried, L.; Genest, F.; Högger, P. Distribution of Constituents and Metabolites of Maritime Pine Bark Extract (Pycnogenol(®)) into Serum, Blood Cells, and Synovial Fluid of Patients with Severe Osteoarthritis: A Randomized Controlled Trial. Nutrients 2017, 9, doi:10.3390/nu9050443.
- Panahi, Y.; Alishiri, G.H.; Parvin, S.; Sahebkar, A. Mitigation of Systemic Oxidative Stress by Curcuminoids in Osteoarthritis: Results of a Randomized Controlled Trial. Journal of dietary supplements 2016, 13, 209-220, doi:10.3109/19390211.2015.1008611.
- Nguyen, C.; Coudeyre, E.; Boutron, I.; Baron, G.; Daste, C.; Lefèvre-Colau, M.M.; Sellam, J.; Zauderer, J.; Berenbaum, F.; Rannou, F. Oral resveratrol in adults with knee osteoarthritis: A randomized placebo-controlled trial (ARTHROL). PLoS medicine 2024, 21, e1004440, doi:10.1371/journal.pmed.1004440.
- Dini, I.; Grumetto, L. Recent Advances in Natural Polyphenol Research. Molecules 2022, 27, doi:10.3390/molecules27248777.
- Hendawy, O.M. Nano-Delivery Systems for Improving Therapeutic Efficiency of Dietary Polyphenols. Alternative therapies in health and medicine 2021, 27, 162-177.
- Kim, H.S.; Quon, M.J.; Kim, J.A. New insights into the mechanisms of polyphenols beyond antioxidant properties; lessons from the green tea polyphenol, epigallocatechin 3-gallate. Redox biology 2014, 2, 187-195, doi:10.1016/j.redox.2013.12.022.
- Ahmed, S. Green tea polyphenol epigallocatechin 3-gallate in arthritis: progress and promise. Arthritis Res Ther 2010, 12, 208, doi:10.1186/ar2982.
- Li, T.; Peng, J.; Li, Q.; Shu, Y.; Zhu, P.; Hao, L. The Mechanism and Role of ADAMTS Protein Family in Osteoarthritis. Biomolecules 2022, 12, doi:10.3390/biom12070959.
- Wong, R.H.X.; Evans, H.M.; Howe, P.R.C. Resveratrol supplementation reduces pain experience by postmenopausal women. Menopause (New York, N.Y.) 2017, 24, 916-922, doi:10.1097/gme.0000000000000861.
- Thaung Zaw, J.J.; Howe, P.R.C.; Wong, R.H.X. Long-term resveratrol supplementation improves pain perception, menopausal symptoms, and overall well-being in postmenopausal women: findings from a 24-month randomized, controlled, crossover trial. Menopause (New York, N.Y.) 2020, 28, 40-49, doi:10.1097/gme.0000000000001643.
- Annaji, M.; Poudel, I.; Boddu, S.H.S.; Arnold, R.D.; Tiwari, A.K.; Babu, R.J. Resveratrol-loaded nanomedicines for cancer applications. Cancer reports (Hoboken, N.J.) 2021, 4, e1353, doi:10.1002/cnr2.1353.
- Scazzocchio, B.; Minghetti, L.; D'Archivio, M. Interaction between Gut Microbiota and Curcumin: A New Key of Understanding for the Health Effects of Curcumin. Nutrients 2020, 12, doi:10.3390/nu12092499.
Round 2
Reviewer 3 Report
Comments and Suggestions for Authors
The authors have addressed my comments and their manuscript has been improved.
There are two minor points that the authors should consider before publication:
1) The last sentence of the abtract should be revised in order to provide clear summary of the aims of this narrative review.
2) L24: compounds instead of "compound".
Author Response
The authors have addressed my comments and their manuscript has been improved.
There are two minor points that the authors should consider before publication:
1) The last sentence of the abtract should be revised in order to provide clear summary of the aims of this narrative review.
Response: Thank you for the suggestion. The sentence is revised as “This narrative review highlighted the nanotherapeutic strategies utilizing bioactive compounds on chondrocyte growth, metabolism, and epigenetic modifications in osteoarthritis amelioration”.
2) L24: compounds instead of "compound".
Response: corrected
